# *Anopheles albimanus* (Diptera: Culicidae) Ensemble Distribution Modeling: Applications for Malaria Elimination

**DOI:** 10.3390/insects13030221

**Published:** 2022-02-22

**Authors:** Charlotte G. Rhodes, Jose R. Loaiza, Luis Mario Romero, José Manuel Gutiérrez Alvarado, Gabriela Delgado, Obdulio Rojas Salas, Melissa Ramírez Rojas, Carlos Aguilar-Avendaño, Ezequías Maynes, José A. Valerín Cordero, Alonso Soto Mora, Chystrie A. Rigg, Aryana Zardkoohi, Monica Prado, Mariel D. Friberg, Luke R. Bergmann, Rodrigo Marín Rodríguez, Gabriel L. Hamer, Luis Fernando Chaves

**Affiliations:** 1Department of Entomology, Texas A&M University, College Station, TX 77843, USA; cgrhodes@tamu.edu (C.G.R.); ghamer@tamu.edu (G.L.H.); 2Instituto de Investigaciones Científicas y Servicios de Alta Tecnología, Ciudad de Panama Apartado Postal 0816-02593, Panama; jloaiza@indicasat.org.pa; 3Programa Centroamericano de Maestría en Entomología, Universidad de Panamá, Ciudad de Panama Apartado Postal 0816-02593, Panama; 4Departamento de Patología, Escuela de Medicina Veterinaria, Universidad Nacional, Heredia Apartado Postal 304-3000, Costa Rica; luis.romero.vega@una.cr; 5Oficina Central de Enlace, Programa Nacional de Manejo Integrado de Vectores, Ministerio de Salud, San José, San Jose Apartado Postal 10123-1000, Costa Rica; baduel@gmail.com (J.M.G.A.); gabriela.delgado@misalud.go.cr (G.D.); carlosm.aguilar@misalud.go.cr (C.A.-A.); rodrigo.marin@misalud.go.cr (R.M.R.); 6Programa Nacional de Manejo Integrado de Vectores, Región Huetar Norte, Ministerio de Salud, Muelle de San Carlos, San Carlos, Alajuela Código 21006, Costa Rica; ogerardors@gmail.com; 7Vigilancia de la Salud, Ministerio de Salud, San José, San Jose Apartado Postal 10123-1000, Costa Rica; melissa.ramirez@misalud.go.cr (M.R.R.); aryanazardkoohi@gmail.com (A.Z.); 8Programa Nacional de Manejo Integrado de Vectores, Región Huetar Caribe, Ministerio de Salud, Sixaola, Talamanca, Limon Código 70402, Costa Rica; ezequias.maynes@misalud.go.cr; 9Coordinación Regional, Programa Nacional de Manejo Integrado de Vectores, Región Pacífico Central, Ministerio de Salud, Puntarenas, Puntarenas Código 60101, Costa Rica; jose.valerin@misalud.go.cr; 10Coordinación Regional, Programa Nacional de Manejo Integrado de Vectores, Región Brunca, Ministerio de Salud, San Isidro del General, Pérez Zeledón, San Jose Código 11901, Costa Rica; alonso.soto@misalud.go.cr; 11Instituto Conmemorativo Gorgas de Estudios de la Salud, Ciudad de Panama Apartado Postal 0816-02593, Panama; chrigg@gorgas.gob.pa; 12Unidad de Investigación en Plasmodium, Centro de Investigación en Enfermedades Tropicales (CIET), Facultad de Microbiología, Universidad de Costa Rica, San Pedro, San Jose Apartado Postal 11501-2060, Costa Rica; monica.pradoporras@ucr.ac.cr; 13Earth System Science Interdisciplinary Center (ESSIC), University of Maryland, College Park, MD 20740, USA; mfriberg@umd.edu; 14NASA Goddard Space Flight Center, Greenbelt, MD 20771, USA; 15Department of Geography, University of British Columbia, Vancouver, BC V6T 1Z2, Canada; luke.bergmann@gmail.com

**Keywords:** gold mining, Costa Rica, *Plasmodium*, *vivax* malaria, productive landscapes, oil palms, pineapples, plantationocene, Schmalhausen’s law

## Abstract

**Simple Summary:**

Costa Rica is near malaria elimination. However, sporadic outbreaks still occur, and while control strategies have been focused on delivering efficient treatments for infected patients, an open question is whether control measures targeting the dominant vector, *Anopheles albimanus*, are appropriately designed given their ecology and distribution. Here, we illustrate the use of an ensemble species distribution model (SDM) as a tool to assess the potential exposure to *An. albimanus* in palm and pineapple plantations, and to also assess the potential involvement of this mosquito vector in transmission foci where entomological surveillance is not feasible. We found that both oil palm and pineapple plantations are very likely to harbor *An. albimanus*. By contrast, environments at the Crucitas open-pit gold mine, the epicenter of malaria transmission in 2018 and 2019, have low suitability for this mosquito species. Our results suggest that medium to high resolution SDMs can be used to plan vector control activities. Finally, we discuss the high suitability of oil palm and pineapple plantations for *An. albimanus* in reference to recently developed social science theory about the Plantationocene.

**Abstract:**

In the absence of entomological information, tools for predicting *Anopheles* spp. presence can help evaluate the entomological risk of malaria transmission. Here, we illustrate how species distribution models (SDM) could quantify potential dominant vector species presence in malaria elimination settings. We fitted a 250 m resolution ensemble SDM for *Anopheles albimanus* Wiedemann. The ensemble SDM included predictions based on seven different algorithms, 110 occurrence records and 70 model projections. SDM covariates included nine environmental variables that were selected based on their importance from an original set of 28 layers that included remotely and spatially interpolated locally measured variables for the land surface of Costa Rica. Goodness of fit for the ensemble SDM was very high, with a minimum AUC of 0.79. We used the resulting ensemble SDM to evaluate differences in habitat suitability (HS) between commercial plantations and surrounding landscapes, finding a higher HS in pineapple and oil palm plantations, suggestive of *An. albimanus* presence, than in surrounding landscapes. The ensemble SDM suggested a low HS for *An. albimanus* at the presumed epicenter of malaria transmission during 2018–2019 in Costa Rica, yet this vector was likely present at the two main towns also affected by the epidemic. Our results illustrate how ensemble SDMs in malaria elimination settings can provide information that could help to improve vector surveillance and control.

## 1. Introduction

Species distribution models (hereafter SDMs) predict species distribution ranges in a space defined by coordinates (hereafter G-space). This concept is commonly confused with environmental niche models (hereafter ENMs), which attempt to depict the species distribution across a series of environmental gradients, or an environmental space (hereafter E-space) [1,2]. These approaches use georeferenced occurrence points and associated environmental information, plus computer algorithms, to generate models of the probabilistic distribution of a species in a E-space that becomes projected into a G-space, while reducing errors regarding species distribution [3,4]. 

Based on both SDMs and ENMs, populations can be conceived as occupying environmental niches that are similar (‘niche similarity’; Peterson et al. [5]) or identical (‘niche equivalency’; Graham et al. [6]). While the first is relevant for testing broad biogeographic and evolutionary hypotheses, the latter is useful for testing the transferability of niche models in space and over relatively short periods of time [7]. In other words, SDMs overcome the limitations of traditional approaches, such as the widely implemented “Extents of Occurrence” [1,8,9], for depicting the spatial range of a species as they are not based on opinions but quantitative relations. SDMs and ENMs can help to forecast trends in biodiversity loss driven by changing environmental conditions to forecast biological invasions and resolve questions about ecological and evolutionary diversification in response to environmental changes [5,10,11,12,13,14]. 

Mosquitoes in the genus *Anopheles* (Diptera: Culicidae) include several vectors of human malaria [15]. Successful malaria control efforts have been largely based on vector reduction [16]. However, anthropogenic changes to the landscape and vector control activities have been accompanied by shifts in major anopheline vector species, which can be assessed employing information from SDMs and ENMs. To take several examples, dominant vector species can adapt and expand into new geographic areas and habitats, become resistant to insecticides, or be displaced by other species, whose genetics and behavior are unknown. Additionally, unidentified anopheline taxa within cryptic species complexes may represent incipient evolutionary units, whose ability to adapt to climate change and transmit *Plasmodium* spp. parasites to humans vary with respect to isolated populations of the same species [17]. All of this evolutionary complexity occurs against a backdrop of environmental alteration driven by human activity, which in turn influences mosquito distribution, species composition and density. The result of these interactions is highly focal and often leads to idiosyncratic malaria transmission patterns that are poorly characterized, and where standard interventions are not well adapted to reduce transmission, as has been observed in the *Anopheles gambiae* complex from Africa [18,19,20]. Hence, there is a critical need to characterize the link between anopheline mosquitoes and the environment, especially in malaria endemic areas where SDMs and ENMs can help to design and implement precise control activities across the suitable habitat of local malaria vectors [21,22,23]. In Mesoamerica, the dominant vector species across the region is *Anopheles albimanus* Wiedemann [24,25]. *An. albimanus* primarily occurs below 500 m [24,25], although it has been observed at higher elevations [26]. This mosquito species is crepuscular, zoophilic and exophagic, with exophilic resting behavior. Larvae and adults have been found over a wide variety of ecological contexts [27]. Prior SDMs for *An. albimanus* in Mesoamerica have been performed at relatively coarsely grained spatial scales between 1 and 8 km [21,27,28], and these efforts have relied on the use of single algorithms, including boosted regression trees [27] and MAXENT [21,28], which have not been evaluated as part of ensembles, which are known to increase the precision and accuracy of SDMs [29].

In Mesoamerica, malaria is still an important vector-borne disease. However, Costa Rica is on the verge of eliminating the disease. This malaria elimination is the result of several control efforts, where elimination has been accelerated following changes in the treatment coupled with mass drug administration campaigns [30,31] and housing quality improvement [32]. Nevertheless, since 2016, malaria cases re-emerged given the transboundary movements of pineapple plantation workers from Nicaragua [30], and illegal gold mining in the Crucitas district of San Carlos county [33]. In the malaria elimination context, SDMs are of great value as they can help to quickly evaluate the potential presence of a vector in an area with malaria transmission, but without entomological information. For example, in Costa Rica, the most recent malaria outbreaks have been controlled during the dry season [31] when it is difficult to collect vector samples [34,35]. Using an SDM, for example, it can be quickly assessed if a dominant vector, such as *An. albimanus*, is, or was, likely to be present in areas with transmission, thus allowing the planning of precise control interventions for the rainy season when the mosquito is more abundant [35]. Similarly, SDMs can be used to determine whether dominant vector species are likely to be present in areas where vulnerable populations migrate for economic reasons. From the perspective of vector control operations, however, SDMs even at 1 km are limited in their ability to help plan precise control operations. Given the current capacities in the national vector control program of Costa Rica [36], fine- and ultra-fine-resolution SDMs will more effectively guide the identification of larval habitats and/or the implementation of insecticide applications following the detection of malaria cases, following current protocols for malaria outbreak mitigation [37].

Here, we use mid-resolution spatial data, at 250 m, where we incorporate several layers derived from remotely sensed and locally measured environmental variables to create an ensemble SDM for *An. albimanus*. This SDM, which combines predictions from an ensemble of several quantitative methodologies, is a robust approximation to the distribution of this major malaria vector, which we use to retrospectively assess the possibility that this vector was present in the transmission foci associated with malaria epidemics in 2018 and 2019 [31,38] and in landscapes used for pineapple production, where some malaria outbreaks have been recurrently observed over recent years.

## 2. Materials and Methods

### 2.1. Mosquito Occurrence Records

We assembled a dataset of *An. albimanus* occurrences with records from collections made largely by the vector control program of the Costa Rican Ministry of Health during surveillance and control activities for vector-borne diseases [39]. Additional occurrence data were obtained from the Global Biodiversity Information Facility (GBIF—https://www.gbif.org/ accessed on 20 February 2021) by searching the terms “*Anopheles albimanus*” for species and “Costa Rica” for country. For the GBIF, we did not restrict the time frame for the search, which allowed us to consider a larger collection of occurrence points. Only occurrences that were georeferenced were selected. We also included records from molecular population genetic studies on *An. albimanus* from southern Mesoamerica, which encompassed extensive mosquito sampling across Costa Rica [28,40,41]. Records from the genetic studies and the vector control program were collected after 2000. Occurrence records used in this study are available online at https://osf.io/acjyg/.

### 2.2. Occurrence Data Quality Control

Occurrence data (Figure 1) were checked for duplicates and records with incomplete location information. When found, these records were removed. Surveillance data commonly suffer from spatial bias for a variety of reasons, including site accessibility and uneven sampling efforts. This clustering of occurrence points can result in the overrepresentation of certain areas and, subsequently, model overfit [42]. As such, occurrence records occurring within 0.5 km of each other were removed using the *spThin* function from the R package “spThin” [43], which is described in [44]. We started with 227 records and ended with 110 occurrence records after thinning the dataset. 

### 2.3. Pseudoabsence Points

While it is possible to fit species distribution models with presence-only data, using presence–absence data has been shown to have superior performance [6]. However, true absence points are rare and particularly difficult to confirm for mobile species [47]. Without true absence points, we rely on an artificial set of absence points, termed pseudoabsence points [47,48]. There are many different strategies for generating these points, so we refer to the suggestions by Barbet-Messin et al. [47], who detail the best sampling method based on the SDM algorithms used. Pseudoabsences were generated using the “SRE” method in the biomod2 package. This method uses a surface range envelope (SRE) model to identify a range of suitable environmental conditions [49]. Pseudoabsence points are then sampled randomly outside of that area, as they are considered to be environmentally dissimilar from the location of presence points.

### 2.4. Environmental Data

We created a multilayer raster that included several variables that have been associated with the occurrence of *An. albimanus*. Table 1 shows all the covariates included in the multilayer raster, whose sources and processing are described in the *Data Sources and Processing* subsection.

### 2.5. Data Sources and Processing

When building our multilayer raster, we used surfaces for the enhanced (EVI) and normalized difference vegetation indices (NDVI) from moderate resolution imaging spectroradiometer (MODIS) images [50,51] as the basis grid. EVI and NDVI are commonly used as vegetation growth proxies [52], with EVI being more appropriate for measuring differences in areas with high canopy and dense vegetation [51]. We also included other raster layers from MODIS, including data for surface temperature [53], as well as a forest/non-forest land use classification based on advanced land observing satellite (ALOS) phased arrayed L-band synthetic aperture radar (PALSAR) images [54]. The PALSAR forest classification is based on identified forests with an area larger than 0.5 ha, with over 10% forest coverage in accordance with the Food and Agriculture Organization (FAO) definition [54]. We also included data for population density from the Gridded Population of the World, Version 4 (GPWv4) with the Population Density Adjusted to Match 2015 Revision of UN WPP Country Totals dataset [55]. Data from the NASADEM_HGT v001 digital elevation model were also included [56].

All raster data were downloaded from Google Earth Engine (GEE) using javascript code available on the GEE website [57]. For MODIS and PALSAR variables that had time series of images, we estimated median, standard deviation, kurtosis and maximum and minimum composite images using the javascript *reducer* function in GEE. The downloaded data were warped, i.e., re-projected and re-sampled [58], using the bicubic spline algorithm and the EVI/NDVI grid as a template, using the command *sf_warp* from the “stars” package of R. We chose the bicubic spline algorithm, given that it has the best performance in terms of precision and accuracy when compared with other algorithms used to resample raster images [59]. The resulting 250 m digital elevation model was further processed to estimate slope, aspect and roughness using the *terrain* function of the “raster” package for R. Slope and aspect were measured in radians. Briefly, slope is the rate of elevational change of the landscape measured in the steepest direction at any point, while the aspect is the direction in which the slope is measured (where 0 is north, π/2 is east, π is south and 3π/2 is west) [60]. Meanwhile, roughness at a given pixel is the largest elevation difference within the set of nine pixels composed by that given (‘focal’) pixel and its eight surrounding neighbor cells in the rectangular raster grid [61]. 

We also included rainfall [62] and temperature [63,64,65] data from the Costa Rican National Meteorological Institute and data about the built environment based on a coupled photogrammetric and cadastral record analysis [45]. These data were vector files [45,62,63,64,65], and were rasterized over the 250 m grid of EVI and NDVI raster layers using the command *sf_rasterize* of the R package “stars”.

All the raster layers were then stacked into a multilayer raster brick with the commands *stack* and *brick* from the “raster” package using R. The resulting multilayer raster, with a resolution of 250 m, is available online at https://osf.io/acjyg/.

### 2.6. Parametric Models, Machine Learning Algorithms and Variable Selection

We employed a parametric model for SDM, the logistic generalized linear model (L-GLM), which can predict the presence and absence of a species based on a linear combination of variables [66]. We also employed the following six machine learning algorithms to produce models that estimate habitat suitability: classification and regression tress (CAT), generalized boosted regression models (GBM) [67], random forests (RF) [68], artificial neural networks (ANN) [68], the multiple adaptive regression splines (MARS) and MAXENT [1,2,3]. GBM and RF are based on the use of CAT, which are computational tools that iteratively find thresholds and other non-linearities in the association of covariates with a response [69]. In the case of GBM, trees are boosted, meaning that simpler trees are combined to improve the accuracy of predictions [67]. In RF, trees are built for resampled datasets in a fashion similar to the one used for building a bootstrap [68,70]. Meanwhile, ANN are models that incorporate non-linearities in the association of variables by using nonlinear functions that combine the information from several variables (called ‘inputs’ in ANN terminology) in layers of neurons, which become combined (‘activated’) to generate a prediction (‘output’) [68]. MARS is a technique that uses spline fitting to find piecewise linear basis functions that accommodate non-linear relationships between the environmental covariates and presence probability of a species [71]. Finally, MAXENT maximizes an entropy function that separates the distribution of environmental variables from pixels where the species has been recorded from the background distribution of the same variables where the studied species has not been sampled, taking into account the constraints derived from environmental conditions [3].

For variable selection, we generated 10 sets of 110 pseudoabsence points, as the process of data cleaning left us with 110 occurrences and machine learning algorithms work best with symmetric datasets [47,68], i.e., with the same number of occurrence and pseudoabsences in this case. Pseudoabsence points were generated using the surface range envelope algorithm described in Section 2.3, where points are chosen at random from areas considered to be environmentally dissimilar from the locations of presence points [47]. We then ran each one of the seven methods mentioned above three times, and each of the three times, we included 100 permutations for each covariate at a time to estimate variable importance, a measurement of the drop in explained variance or prediction accuracy. Based on this preliminary analysis, we chose all variables whose importance was above 5%. Once a reduced number of covariates was selected, we generated an additional pseudoabsence dataset of 110 locations, which was run ten times with each one of the seven models, including 100 permutations for each covariate to assess their importance. We used the resulting values to generate an ensemble SDM map that weighted all the resulting 70 projected predictions for *An. albimanus* habitat suitability, using the ROC value from each individual model.

Evaluation strip plots were employed to visualize the probability of occurrence response curve for each covariate in the final model. The strip plots are generated by producing a prediction from a model using a new dataset in which only one variable is allowed to vary in a sequence between the minimum and maximum, while the other variables are fixed at their median values [72].

All analyses were completed using the biomod2 package [49,73] for R [74]. This package was selected for its ability to incorporate several techniques and for its reproducibility. Five folds were created with 10 repetitions, and each data partition contained approximately the same number of presence points. The *BIOMOD_Modeling* function was used for model generation, and a total of 280 individual models were fit for variable selection, and 70 additional models were fit to generate the SDM map. As mentioned above, models were evaluated using k-fold cross-validation, and the evaluation statistics returned were area under the curve (AUC) and the true skill statistic (TSS). The AUC statistic estimates the model’s likelihood to correctly differentiate between presence and absence locations, with a value of 0.5 suggesting that model performance is no better than random chance. The TSS statistic works similarly and is equal to the sum of model sensitivity and specificity minus one. A final ensemble model was fit to include all models with an AUC score of 0.7 or higher, and each model was weighted proportionally to its AUC score. We only consider AUC scores of 0.7 and above, as they are considered to demonstrate high model performance [75]. 

### 2.7. Applications for SDMs in the Context of Malaria Elimination

We used the resulting ensemble SDM for *An. albimanus* to investigate its probable presence in the productive landscapes of Costa Rica, i.e., areas with plantations of commodity crops for export. We also retrospectively evaluated if *An. albimanus* was likely to be present in the 2018–2019 malaria outbreak associated with open-pit gold mining in Crucitas [31].

#### 2.7.1. Background Information about Productive Landscapes in Costa Rica

In Costa Rica, oil palm has been a major commodity crop for export since the 1940s, with the country producing around 190,000 metric tons of crude oil per year [76,77]. As of 2019, Costa Rica has 73,900 hectares in oil palm plantations [78]. Another common commodity crop is pineapple. An interesting feature of pineapple plantations in Costa Rica is that their total area has been increasing in recent years, rising from approximately 13,300 ha in 2000 to 65,400 ha in 2019 [79]. This substantial shift in land use, central to the development of the northern border in the country, has occurred as pineapple has become a major commodity crop for markets in Europe and North America [80]. Currently, Costa Rica is the main global producer of pineapples, reporting revenues close to USD one billion per year [81]. Figure 2 shows the location and extent of palm oil (Figure 2A) and pineapple (Figure 2B), which are present, respectively, in 80 and 51 districts out of 487 districts in the country, according to estimates for 2019.

#### 2.7.2. *Anopheles albimanus* in Productive Landscapes of Costa Rica

Oil palm plantations (Figure 2A) are common throughout Costa Rica, but have not been associated with malaria outbreaks. A major characteristic of oil palm plantations is the use of residues as compost, which is managed in a way that reduces the abundance of *Anopheles* spp. mosquitoes [82,83]. By contrast, a common feature of many recent malaria outbreaks in Costa Rica has been their apparent association with pineapple plantations [38]. In the context of malaria elimination, it is important to understand the entomological risk associated with the landscapes used to grow these two major commodity crops. As a proof of concept, we compared *An. albimanus* habitat suitability, measured as a probability, in land used for palm (Figure 2A) and pineapple (Figure 2B) plantations with that of the remaining land in the plantation districts. Based on estimates for *An. albimanus* dispersal, which has been recorded as occurring in distances of up to 3 km [84,85], we also compared the suitability in the plantations plus buffers of 1, 2 and 3 km with that of the remaining plantation-surrounding land in the plantation districts (Figure 3).

#### 2.7.3. *Anopheles albimanus* in the Crucitas Outbreak of 2018–2019

We also used the resulting *An. albimanus* ensemble distribution model to investigate if the 2018–2019 malaria outbreak observed in locations within the Cutris and Pocosol districts of San Carlos county (Canton San Carlos in Spanish) [31] occurred in areas with high suitability for *An. albimanus*. We evaluated the suitability of *An. albimanus* in circular areas of 3, 5 and 7 km from the population center in the Crucitas open-pit gold mine and the towns of Llano Verde and Boca Arenal.

## 3. Results

We obtained 227 occurrence points for *An. albimanus*. Following data cleaning, 110 records remained, most of which were generally located around the perimeter of Costa Rica (Figure 1). Figure 4 shows the correlations of the different covariates at the pixels with points where *An. albimanus* has been collected. 

Pearson’s correlation matrix (Figure 4) reveals clusters of high correlations (absolute value of Pearson’s r > 0.6). Particularly for the MODIS-based temperature variables, station-based temperature, EVI and NDVI, measures of variability (kurtosis and standard deviation), as well as minimum, maximum and average values. These patterns of association called for a process of variable selection based on variable importance, whose results are presented in Figure 5. Of the twenty-eight environmental variables initially investigated, only nine were selected for inclusion in the final ensemble model. During model selection, variable importance was over 5% for only nine environmental covariates (Figure 5A). Elevation and standard deviation of NDVI were the two most important variables, being prescribed 29% and 16% variable importance, respectively. Of the nine remaining, the most important environmental covariates were elevation, minimum temperature and standard deviation of NDVI (Figure 5B). Using seven different algorithms, a total of 70 models were generated to estimate habitat suitability. AUC and TSS scores for each algorithm, reported in Table 2, indicate universally strong model performance.

RF and MAXENT demonstrated the strongest performance with AUC scores of 0.92 ± 0.04 and TSS scores of 0.76 ± 0.10. Classification and regression trees demonstrated good performance, but comparatively were the worst performing algorithm (AUC = 0.79 ± 0.07, TSS = 0.58 ± 0.13). From the 70 models created, a single final weighted ensemble model was created (Figure 6).

Habitat suitability ranged across Costa Rica from 0 to 1, with a score of 1 representing a habitat where *An. albimanus* should be present. There is some variation in the classification of these scores, but we consider scores ranging from 0 to 0.3 to have poor suitability, 0.3 to 0.5 to be moderately suitable, 0.5 to 0.7 to have good habitat suitability and 0.7 to 1 to be a highly suitable environment [75]. The ensemble model estimates the lowest suitability for *An. albimanus* to be across the central mountain range of the country, the Cordillera Central, with areas of higher suitability patchily distributed throughout the country lowlands and concentrated along the perimeter of the country (Figure 6). Areas of particularly high suitability (probability > 0.7) include a patch in the southeastern portion of the country in the Pacific basin, including regions bordering with Panamá, two strips along the southern border and a patch just below the middle of the northern border; all these patches are concentrated in the Atlantic basin of Costa Rica. The contribution of the different environmental variables in shaping the ensemble SDM for *An. albimanus* can be seen in Figure 7.

The probability of occurrence decreases as elevation increases, but drops dramatically once elevation exceeds 1000 m according to all the algorithms studied (Figure 7A). The probability of occurrence is not largely affected by NDVI (Figure 7B) or maximum NDVI (Figure 7C), with probabilities just decreasing for extreme large values. For most model types, a standard deviation of NDVI (Figure 7D) is negatively correlated with probability of occurrence, especially once the value is greater than 0.2. However, very little correlation is observed when the ANN and classification trees algorithms were used. Suitability only changed with population density (Figure 7E) for the L-GLM, where it decreased for extremely high densities. Rain (Figure 7F) did not have an impact on suitability for most models, the only exception being the L-GLM and MAXENT, where suitability monotonically decreased as rainfall increased. Meanwhile a non-monotonic decrease in habitat suitability was observed with ANN as rainfall increased. Roughness increases (Figure 7G) where mainly associated with a monotonic decrease in *An. albimanus* suitability, according to MAXENT. The kurtosis of MODIS-based land surface temperature (Figure 7H) suggests that more platykurtic environments, i.e., those with low kurtosis, increased habitat suitability for *An. albimanus*, with the exception of ANN and RF, which where insensitive to changes in kurtosis. Generally, minimum temperature (Figure 7I) has a positive correlation with the probability of occurrence. However, when temperatures begin to increase past 30 degrees, the correlation becomes negative. Only when these variables reach the upper values of their distribution, do we tend to see a decline in the probability of occurrence.

A comparison between the presence of plantations (Figure 2) and habitat suitability for *An. albimamus* (Figure 6) suggests that there is a strong relationship between the two. This correlation is further confirmed by a comparison of habitat suitability in plantations and non-plantation landscapes (Figure 8). 

For both oil palm (Figure 8A–D) and pineapple plantations (Figure 8E–H), the habitat suitability is significantly greater compared to non-plantation areas in plantation districts. The addition of spatial buffers decreases the difference in habitat suitability between plantation and non-plantation landscapes, but at all spatial buffers, habitat suitability is still overwhelmingly greater around both oil palm and pineapple plantations.

When considering the relationship between habitat suitability and the 2018–2019 malaria outbreak, areas with a suitability over 0.7 are seen in the two outbreak districts examined (Figure 9A). The majority of Crucitas has low habitat suitability, but near the southwestern portion of the district, there is a small region of moderate-to-high suitability (Figure 9B). Suitability is very low, around 10% at 3km, and increases as the radius of the area considered increases from the center of the open-pit gold mine, up to 17% at 7 km (Table 3). Ranges of habitat distribution in Llano Verde are patchy (Figure 9C); however, suitability was slightly above 30% across the buffer (Table 3). Boca Arenal has the most consistently high habitat suitability (Figure 9D), with 70% suitability at 3 km, a value that slightly decreased as the buffer radius increased (Table 3). 

## 4. Discussion

Our results suggest that mid-to-high spatial resolution SDMs could become an essential part of the toolkit used in routine vector control program operations. The first interesting feature of our ensemble model was that seven out of the nine variables employed as covariates to produce the SDMs were from remotely sensed data, which has the potential to streamline SDMs that automatically update predictions as new data become available. All the remotely sensed variables that were in the models used for the ensemble have been reported to be associated with *An. albimanus* presence and abundance. The most important variable was elevation, accounting for 25 to 30% of variability in the SDMs, which had a robust pattern across the different models, with all models suggesting that habitat suitability decreased as elevation increased. This pattern can reflect two things. Firstly, it reflects the known association between elevation and temperature, where the latter increases and the former decreases [20]. Secondly, it could also be related to landscape features, as lowlands have concentrations of wetlands and other habitats that are known to harbor large densities of *An. albimanus* [87,88,89,90]. This suggestion is also supported by the decrease in suitability with roughness for a subset of the models and increasing minimum temperature. *An. albimanus* population dynamics studies have shown a negative relationship between increased rainfall and abundance [34,35], which could in turn translate into those areas consistently receiving more rainfall as being less likely to be suitable for *An. albimanus*, as observed in the models where rainfall was an important covariate. Similarly, suitability decreases at very high NDVI values and reflects the ecology of *An. albimanus*, as the species prefers sunlit habitats [27,87], and similar patterns of occurrence, where the mosquito is not present in land with near-saturation NDVI values or the dense vegetation cover it is associated with, have been observed elsewhere in Mesoamerica [88,91,92,93,94,95]. Increasing population density per km^2^ was only associated with a decrease in suitability for L-GLM models, and this result could reflect the lack of larval habitats for *An. albimanus* in more urbanized landscapes, while also highlighting the importance of making SDMs with ensembles of models, since the possibility to incorporate information from different covariates increases, as they can be incorporated with different functional forms [29,73].

A novel result from our study is the association of habitat suitability with measures of higher order of variability in the environmental variables. Across most models, habitat suitability decreased as the variance of NDVI increased, suggesting that *An. albimanus* occurrence is associated with landscapes that are relatively stable in terms of vegetation change. However, the association with MODIS-based temperature kurtosis indicates that habitats with low kurtosis, i.e., having platykurtic distributions, are where covariates are relatively more variable towards the mean than the extremes [96,97]. A significant association with kurtosis and the SD of environmental variables illustrates how *An. albimanus* distribution is sensitive both to average environmental conditions and their patterns of variability, following Schmalhausen’s law [98], the ecological principle that indicates that sensitivity to different environmental variables could increase as the limits of tolerance to any environmental variable are reached, and that organisms are, therefore, sensitive to changes in the different statistical moments of environmental factors shaping their abundance and distribution [20].

The resulting ensemble SDM has an increased resolution when compared with previous efforts that have generated SDMs for the territory of Costa Rica, and ranged between 1 and 8 km [21,27,28]. This increased resolution, at 250 m, could potentially help optimize field activities by highlighting areas needing entomological surveillance. We illustrate this with the two applications that we developed in this study.

In the first application, we compared the *An. albimanus* habitat suitability in plantations of two major commodity crops in Costa Rica. One has been long established, as is the case for oil palms, and the other is an emerging global commodity crop, where Costa Rica is the main global producer [81], as is the case with pineapples. Interestingly, in both cases, the suitability was increased in the plantations when compared with surrounding areas in districts where plantations are located, a result robust to different assumptions about the area of influence of a plantation, assuming a maximum 3 km dispersal for *An. albimanus* [84]. The environmental homogenization driven by monocultures [99,100] such as oil palm and pineapple plantations provides new habitats for *An. albimanus* to thrive in Costa Rica’s lowlands. An open canopy for sunlit man-made irrigation habitats, ground wheel tracks created by vehicles used in commodity crop production and transportation and microclimatic conditions that could enhance mosquito reproduction and survival are a few examples of conditions that can enhance the fitness of *An. albimanus* in light of what we know about its life history and ecology [87]. Nevertheless, little research has been conducted into *An. albimanus* in oil palm and pineapple plantations as a malaria transmission risk factor. 

Furthermore, some recent malaria outbreaks have been associated with pineapple plantations [30,31], and we are not aware of entomological studies in such plantations. In contrast, for oil palm plantations, a few entomological studies have shown that such plantations lead to decreases in anopheline abundance when compared with the previous vegetation type [82,83]. These studies suggest that decreases in anopheline mosquitoes might be related to water management practices that reduce the likelihood of *Anopheles* spp. larval habitats [82,83]. Although some ecological conditions are present for the development of *An. albimanus*, little malaria has been observed in the area of Costa Rica dominated by oil palm plantations, and part of that could be related to a decrease in the entomological risk of transmission. However, part of the difference could also be related to historical patterns in the social development of southwestern Costa Rica. This area, where oil palm plantations are concentrated, has historically seen low malaria transmission [32,38]. The development and settlement of oil palm plantations occurred when the Costa Rican state was engaged in the development of a social welfare state where plantation workers had access to decent housing, i.e., housing built of materials, and with characteristics, that reduce exposure to mosquito bites [32] and access to basic healthcare and other services [101], which seems privileged when compared with current working conditions at pineapple plantations in northern Costa Rica, which have been promoted by a neoliberal economic context [80] whose tendencies to impose austerity in public health investment have been associated with the emergence and re-emergence of vector-borne diseases globally [99]. This possibility is further re-enforced when looking at historical malaria patterns in Costa Rica. The disease used to be concentrated in the Caribbean basin when the United Fruit Company (UFCo) started a commodity crop economy with banana plantations [38] that imposed extremely detrimental working and living conditions for the workers, as exposed in the literary work of Carlos Luis Fallas [102] and Joaquin Gutiérrez [103], who vividly described the heavy toll of tropical diseases on plantation workers. Beyond these literary depictions, the large numbers of cases and deaths were even recorded by the UFCo itself [104]. Recent developments in social scientific theories have described this association between plantations and depauperated social, environmental and health conditions as the Plantationocene [105]. Our results suggest that incorporating the model of social and economic relations imposed by the Plantationocene might be key to understanding the differences between oil palm and pineapple plantations in terms of the generation of malaria outbreaks in Costa Rica. Beyond this study, examining the role of the Plantationocene in generating spatial patterns in diseases is an important research question to prevent the emergence of infectious diseases beyond the need of ecological research on mosquito abundance and infection in both types of plantations.

Our second application asked if habitat suitability for *An. albimanus* implied that this was the main vector during the 2018–2019 malaria outbreak associated with illegal open pit-gold mining in Crucitas [31]. Our results suggest that the likelihood of *Anopheles albimanus* being present at the mine itself was very low, with a suitability of only 10%. And this result unlikely reflects “out of date” land cover data, as we considered environmental information that overlapped with the time of the Crucitas malaria outbreak, having ourselves processed a time series of satellite-derived images that included images taken as the outbreak was happening. However, the mosquito was likely present at the two main towns also affected by the epidemic. This result highlights that as malaria cases become rarer following elimination efforts, additional secondary vector species of *Anopheles* could be responsible for transmission. In principle, this calls for improved entomological surveillance in the field. In that sense, we think that SDMs could be extremely useful to prioritize areas needing entomological surveillance following malaria outbreaks, especially as *An. albimanus* has been observed in mining areas of Colombia [24], but our model does not indicate that the Crucitas environment is suitable for its development. This notwithstanding, other vector species might be potentially present in the Crucitas open-pit gold mine, with possibilities including *Anopheles vestitipennis* Dyar & Knab and *Anopheles punctimacula* Dyar & Knab, as these species thrive in recently disturbed environments and commonly co-occur with *An. albimanus* in Mesoamerica [106,107]. Similarly, *Anopheles darlingi* Root, has been predicted to be present in the area with SDMs [108] and has been reported in Panamá [109], but has not yet been detected in Costa Rica.

Finally, this study clearly shows the advantages of developing finely grained SDMs for vectors, as they produce information that could help guide research, surveillance and control efforts for vector-borne diseases as part of efforts for more precise [110,111] public health practice.

## Figures and Tables

**Figure 1 insects-13-00221-f001:**
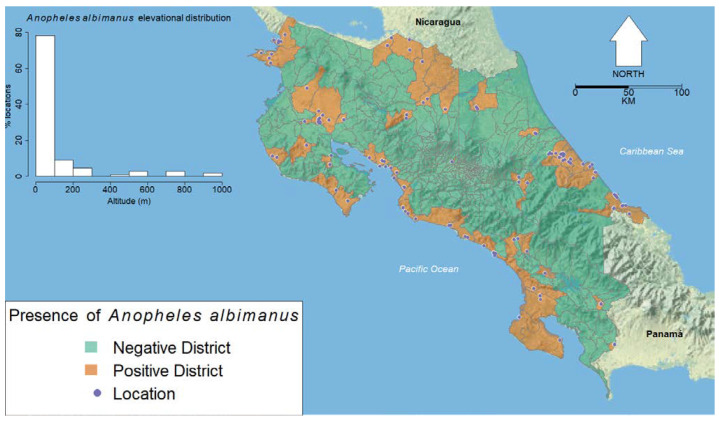
Map showing the distribution of *Anopheles albimanus* record locations; divisions indicate the districts of Costa Rica. The inset histogram shows the distribution of elevations from locations where *An. albimanus* has been sampled. The inset legend indicates symbols and color coding for the presence of *An. albimanus.* The vector file for the districts of Costa Rica is from Costa Rica’s National Geographical Institute [45]. The map used a public domain map from the US National Park Service as its base [46]. *An. albimanus* has been recorded in all seven provinces of Costa Rica, in 28 out of 83 counties and in 55 out of 487 districts.

**Figure 2 insects-13-00221-f002:**
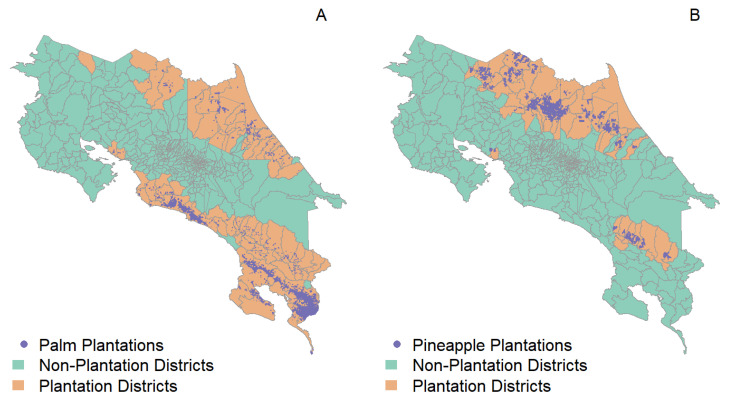
Major plantation landscapes of Costa Rica for (**A**) oil palm and (**B**) pineapple. In both panels, the inset legends indicate the location of the plantations, and districts are colored according to the presence of plantations. Estimates are for 2019 and based on estimates from the PRIAS lab at the Centro Nacional de Alta Tecnología, CENAT [78,79].

**Figure 3 insects-13-00221-f003:**
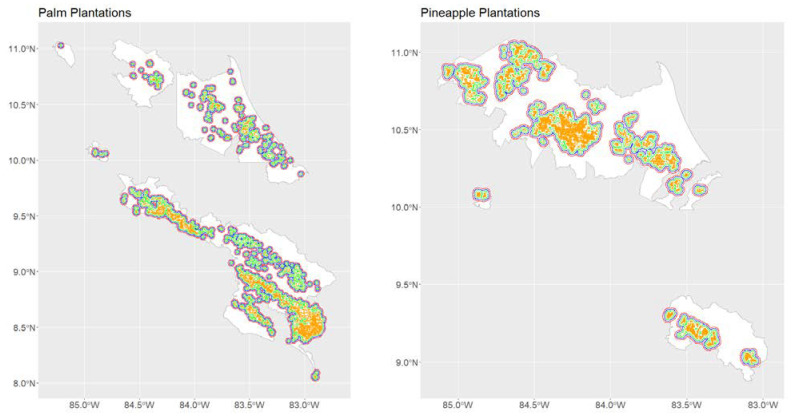
Plantation districts (in white) and plantations (in orange), and the different spatial buffers, 1 km (yellow), 2 km (blue) and 3 km (red), used for comparing *Anopheles albimanus* habitat suitability between plantations and surrounding land in plantation districts. As indicated by the inset titles, the left panel shows the spatial buffers for oil palm plantations, and the right panel shows those for pineapple plantations.

**Figure 4 insects-13-00221-f004:**
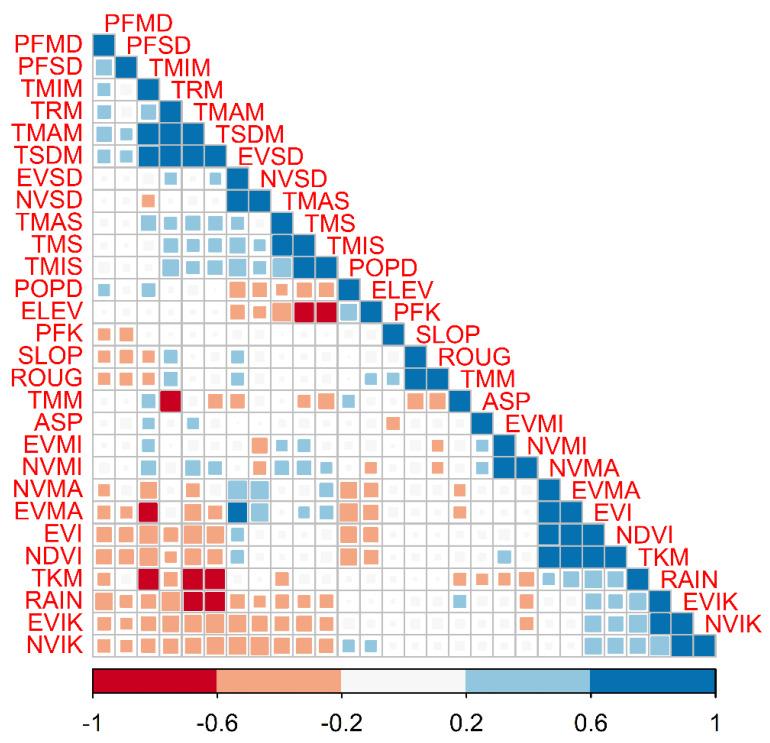
Pairwise Pearson’s correlation between environmental variables at the occurrence points for *Anopheles albimanus* in Costa Rica. Correlations have been clustered to ease the visualization of groups of highly correlated variables.

**Figure 5 insects-13-00221-f005:**
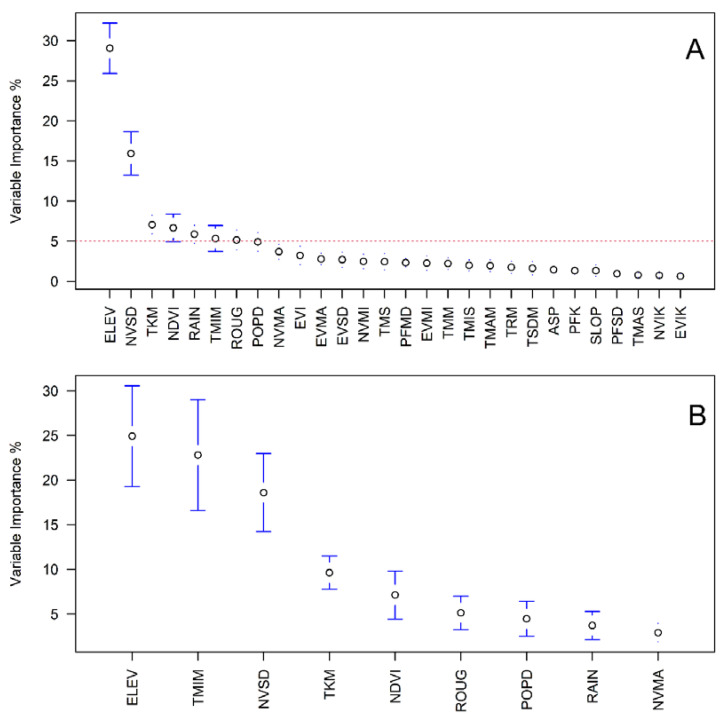
Variable importance in species distribution models of *Anopheles albimanus* (**A**) used for model selection (the dashed line indicates the 5% importance threshold) and (**B**) in the models used for the final ensemble prediction.

**Figure 6 insects-13-00221-f006:**
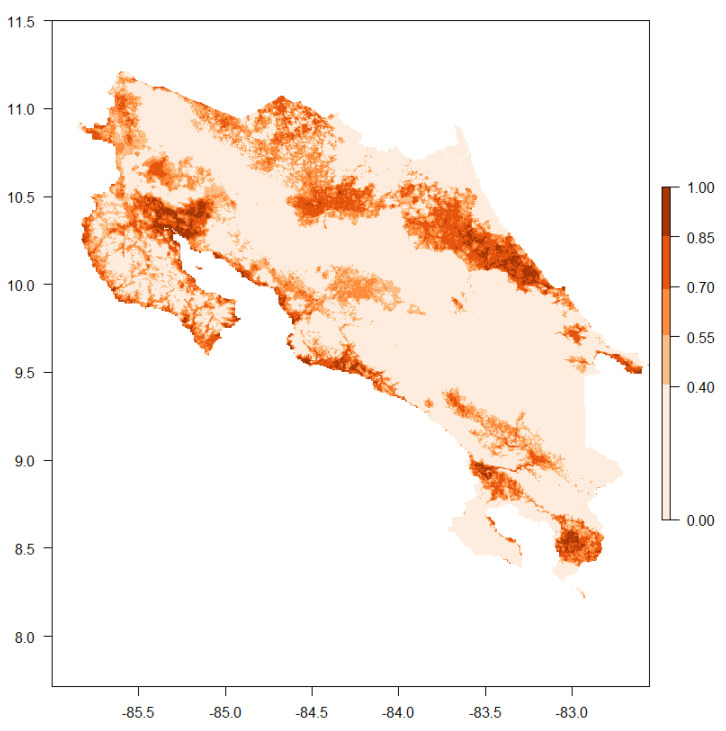
Ensemble distribution model for *Anopheles albimanus* in Costa Rica. Color indicates habitat suitability measured as a probability from 0 to 1 as presented in the legend; the Y axis is the latitude, and the X axis is the longitude. The AUC (mean ± SD) for this ensemble model was 0.983 ± 0.001 and the TSS (mean ± SD) was 0.833 ± 0.007, outperforming all individual models presented in Table 2. These raster results are available at https://osf.io/acjyg/.

**Figure 7 insects-13-00221-f007:**
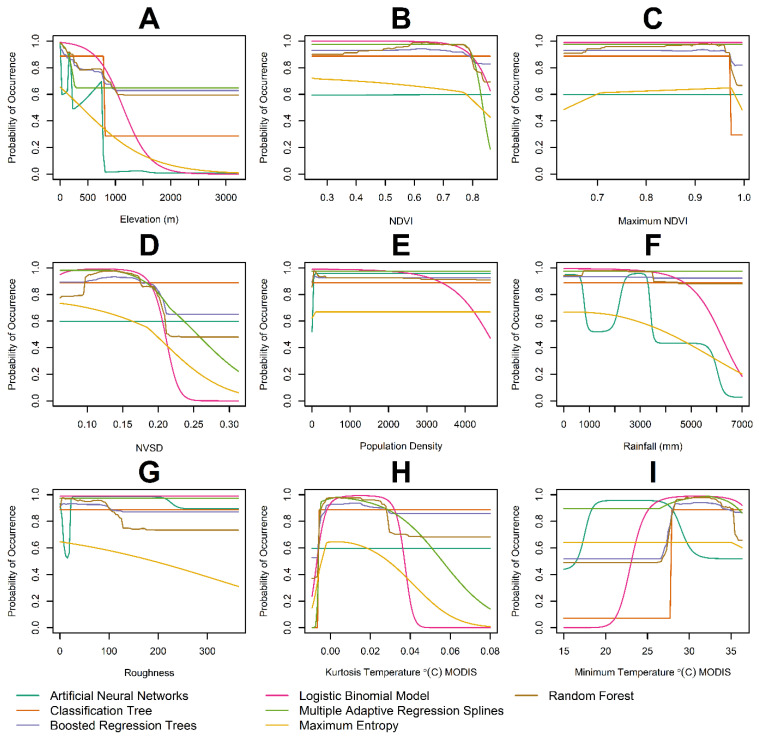
Strip plots of the predicted probability of *Anopheles albimanus* occurrence as function of the covariates considered in the ensemble of best models. In the plots, each line represents the mean for the replicated runs of each model. Model methodologies are color-coded (see bottom of the figure). Covariates included (**A**) elevation, (**B**) NDVI, (**C**) maximum NDVI, (**D**) SD of NDVI, (**E**) population density per km^2^, (**F**) rainfall measured in mm, (**G**) landscape roughness, (**H**) MODIS-based temperature kurtosis and (**I**) MODIS-based minimum temperature in °C.

**Figure 8 insects-13-00221-f008:**
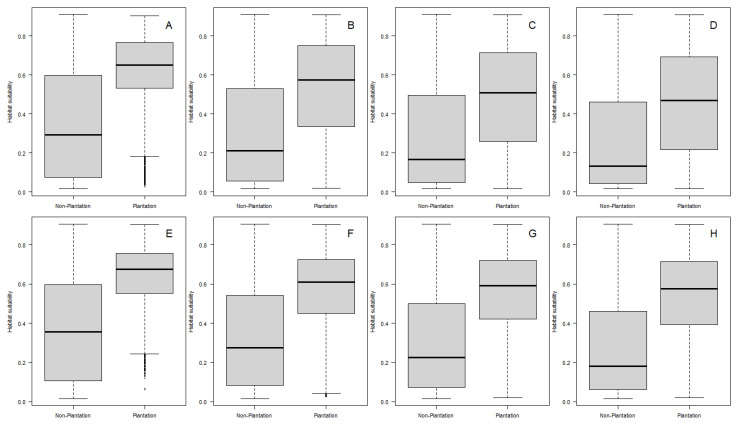
Boxplots comparing *Anopheles albimanus* habitat suitability in plantations and non-plantation landscapes from plantation districts for oil palms (**A**) without a spatial buffer, *t* = 170.01, df = 13,879, *p*-value < 2.2 × 10^−16^, (**B**) with a 1 km spatial buffer, *t* = 200.54, df = 99,252, *p*-value < 2.2 × 10^−16^, (**C**) with a 2 km spatial buffer, *t* = 204.02, df = 209,115, *p*-value < 2.2 × 10^−16^ and (**D**) with a 3 km spatial buffer, *t* = 210.34, df = 298,572, *p*-value < 2.2 × 10^−16^; and for pineapples (**E**) without a spatial buffer, *t* = 177.12, df = 14,391, *p*-value < 2.2 × 10^−16^, (**F**) with a 1 km spatial buffer, *t* = 242.18, df = 107,952, *p*-value < 2.2 × 10^−16^, (**G**) with a 2 km spatial buffer, *t* = 262.72, df = 172,789, *p*-value < 2.2 × 10^−16^ and (**H**) with a 3 km spatial buffer, *t* = 278.23, df = 208,638, *p*-value < 2.2 × 10^−16^. In all panels, the boxplots show the median for the distribution of pixel values, while the boxes represent the 25th and 75th percentiles of the data. For each boxplot comparison, we report *t* values for two-sample Welch’s *t* tests, a statistic used to compare means between two groups, in this case plantation vs. non-plantation pixels. We chose Welch’s *t* test for the comparison because it accounts for heterogeneous variances in the compared groups [86].

**Figure 9 insects-13-00221-f009:**
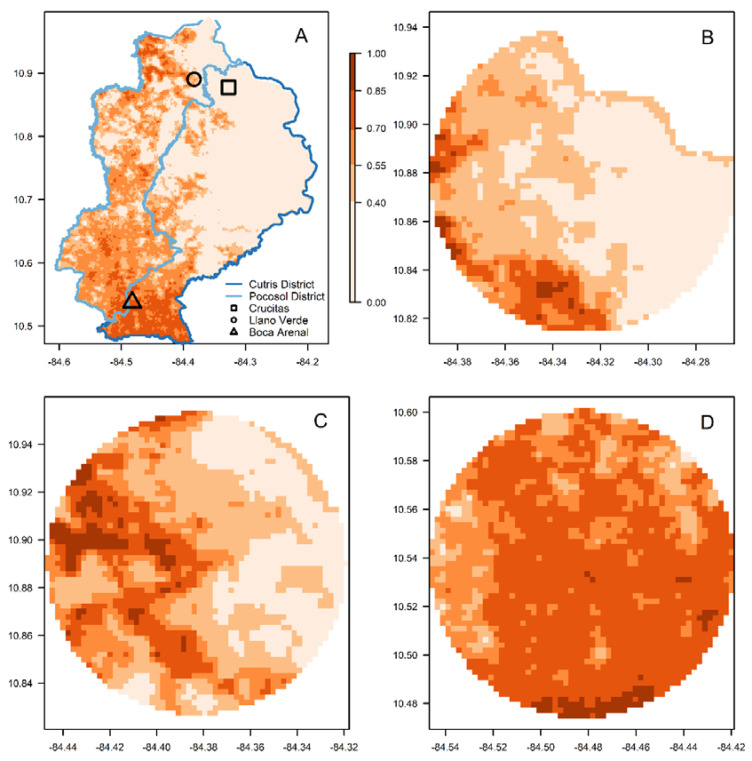
Ensemble distribution model for *Anopheles albimanus* in (**A**) Cutris and Pocosol districts, where symbols indicate the locations associated with the 2018–2019 Crucitas malaria outbreak (for details, refer to the inset legend), and in (**B**) Crucitas, (**C**) Llano Verde and (**D**) Boca Arenal. In all panels, color indicates habitat suitability quantified by probabilities from 0 to 1, as presented in the legend of panel (**A**). In all panels, the Y axis is the latitude and the X axis is the longitude. In all panels, each pixel is a 250 m square, and in panels B, C and D, the circular areas have a 7 km radius.

**Table 1 insects-13-00221-t001:** Environmental covariates considered for *Anopheles albimanus* species distribution modeling in Costa Rica. * Geological feature, ** vector files.

Covariate	Raster Original Spatial Resolution (Covariate Units)	Frequency (Period Sampled)	Derived Layers (Abbreviation)
MODIS—Enhanced Vegetation Index (EVI)	250 m (Adimensional Ratio)	16 days (2000-02-24 to 2019-12-31)	Standard Deviation, SD (EVSD) Kurtosis (EVIK) Maximum (EVMA) Minimum (EVMI) Median (EVI)
MODIS—Normalized Difference Vegetation Index (NDVI)	250 m (Adimensional Ratio)	16 days (2000-02-24 to 2019-12-31)	SD (NVSD) Kurtosis (NVIK) Maximum (NVMA) Minimum (NVMI) Median (NDVI)
MODIS—Land Surface Temperature	1000 m (° Kelvin)	Daily (2000-02-24 to 2019-12-31)	SD (TSDM) Kurtosis (TKM) Maximum (TMAM) Minimum (TMIM) Range (TRM) Median (TMM)
PALSAR—Forest/Non-Forest	25 m (1 = Forest, 2 = Non-forest, 3 = Water)	Annual (2007–2019)	Mode (PFC) SD (PFSD) Kurtosis (PFK)
NASA—Digital Elevation Model	30 m (Meters Above Sea Level)	2000 s *	Elevation (ELEV) Aspect (ASP) Roughness (ROUG) Slope (SLOP)
GPWv4—Population Density	1000 m (Population Density)	2015	Population Density (POPD)
INM—Rainfall	1:5000 (mm) **	Annual average based on daily records (1963-01-01 2013-12-31)	Rainfall (RAIN)
INM—Temperature	1:5000 (°C) **	Annual average based on daily records (1963-01-01 2013-12-31)	Mean (TMS) Minimum (TMIS) Maximum (TMAS)

**Table 2 insects-13-00221-t002:** Mean area under curve (AUC) and true skill statistic (TSS) values, with standard deviation (±SD), based on the 10 model repetitions per algorithm that were used to build the ensemble distribution model for *Anopheles albimanus* in Costa Rica. Abbreviations: L-GLM, logistic generalized linear model; MARS, multiple adaptive regression spline; CAT, classification and regression trees; RF, random forest; GBM, generalized boosting model; ANN, artificial neural networks; MAXENT, maximum entropy.

Algorithm	AUC	TSS
L-GLM	0.91 ± 0.05	0.75 ± 0.11
MARS	0.89 ± 0.04	0.68 ± 0.09
CAT	0.79 ± 0.07	0.58 ± 0.13
RF	0.92 ± 0.04	0.76 ± 0.10
GBM	0.91 ± 0.05	0.72 ± 0.10
ANN	0.85 ± 0.07	0.63 ± 0.12
MAXENT	0.92 ± 0.04	0.76 ± 0.10

**Table 3 insects-13-00221-t003:** *Anopheles albimanus* mean habitat suitability (HS) probability around the three locations of the 2018–2019 malaria outbreak associated with illegal open-pit gold mining in Crucitas, Costa Rica.

Location	Spatial Buffer Radius (HS Mean ± S.D.)		
	3 km	5 km	7 km
Crucitas	0.10 ± 0.04	0.12 ± 0.09	0.17 ± 0.15
Llano Verde	0.33 ± 0.18	0.33 ± 0.19	0.31 ± 0.19
Boca Arenal	0.70 ± 0.05	0.69 ± 0.06	0.67 ± 0.09

## Data Availability

All data are freely available at the following repository: doi:10.17605/OSF.IO/ACJYG.

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
