# Peer review of "Anopheles albimanus (Diptera: Culicidae) Ensemble Distribution Modeling: Applications for Malaria Elimination"

_insects, 2022, doi:10.3390/insects13030221_

Round 1
Reviewer 1 Report
This manuscript explored the application of species distribution models in malaria elimination settings. Specifically, authors first established an ensemble specie distribution model(SDM) with Anopheles albimanus past presence/ peseudoabsence data as the response and 28 environmental variables as the predictors; and then used the best fitted model as the standard to evaluate the habitat suitability (HS) for the presence of this species in commercial plantations and areas that had malaria outbreaks in 2018 and 2019; authors reported a higher HS in commercial plantations than in their surrounding landscapes, but a low HS in areas that were the presumed epicenter of malaria transmission.
The manuscript has provided enriching details and the design of study seems rational. Overall, I believe the authors' results have some implications for the surveillance and control of vectors. However, I do find several elements of the manuscript difficult to interpret in their current form.
Major issues
- Some environmental variables included in the ensemble SDM showed very high or low values, which may affect the generality of the modeling for other areas. For example, the rainfall ranges from 1000 to 5000 mm (Fig. 7F), while many areas in the world have a rainfall lower than 1000 mm. Rainfall may have a positive impact on mosquito population dynamics in many areas because it increases the humidity and creates more habitats for larvae. However, here authors reported a negative impact of rainfall on mosquito occurrence. This discrepancy could be due to the distinct difference between the ranges of rainfall. So I wonder how the modeling will be affected when rainfall is lower than 1000 mm, a range might seem reasonable for many areas of the world. And same consideration for population density (1000-4000 people per km2) and temperature (15-35 ℃), what the modeling will be changed if the population density is much higher and temperature is lower, which are likely scenarios for many areas. This additional considerations and improvements may help improve the generality of the modeling.
- In one the authors’ major finding, HS is significantly higher in commercial plantations than in their surroundings. However, the values of HS for plantations are mostly around 0.4-0.6. If the range of HS is between 0 and 1, does this mean HS in plantations is pretty neutral, that is, mosquito occurrence is no better than other areas?
- In the method section, how authors applied the standard/best-fitted SDM to evaluate the HS of plantations and malaria transmission areas are unclear. Did you collect data of the same environmental variables from the plantations and malaria transmission areas and then substitute them into the established regression between mosquito presence/absence and environmental variables to get the new presence/absence results? More details including the environmental data collected from those two studied areas are needed for clarity.
- Low HS in malaria transmission areas during 2018 and 2019 and relatively neutral HS in plantations may indicate that some important predictors are missing. Have you considered the number of water puddles, distance to water body, and intensity of mosquito intervention in the studied areas? Adding these predictor variables may enhance the explanatory power of the modeling.
Minor issues
Figure 1: Some positive areas are represented by one record location only, whereas other positive areas have multiple location points. Did this cause any overrepresenting/underrepresenting bias in the modeling, how did you balance the potential bias?
L199: what about the fine resolution images? Given that habitats of mosquitoes, especially larvae are very small, using fine resolution images may help improve the models.
L292: Why choosing an AUC score of 0.7 as the inclusion criterium? “was fit” or “was fitted”?
Figure 5: why NVMA was included in the lower model given the upper model showed its importance smaller than 5%?
L412: rainfall may have a significant impact on mosquito occurrence when it is limiting the mosquito life history. As mentioned above, consider adding data of lower rainfall (< 1000 mm) to enhance the generality of modeling.
L456: A low HS in malaria transmission areas deserves more details and discussion, including the possibility of missing important predictors, such as number of water puddles and intensity of mosquito interventions.
Author Response
We thank the reviewer for checking the manuscript. In the following lines we itemize replies to each one of the major and minor comments using the same numbers.
Major comments
1) The reviewer mentions many areas in the world, but it is important to remember that this model was created specifically for Costa Rica. While we could apply this niche model to other countries/scenarios it would be careless to do so without some revisions to the model. When we broaden the spatial extent of an SDM and generalize the analysis, we may lose a lot of the local variation in environmental factors and ultimately have poorer model performance. Essentially, the model will not be able to pick up local biotic interactions. Our model uses very fine spatial resolution and limits the analysis to Costa Rica to get a very specific view of the habitat requirements here. So while there is large variation in some variables within Costa Rica, we are confident that we have captured the overall dynamics and trends pertinent to the habitat suitability of An. albimanus in Costa Rica.
2) For many statistics that range from 0 to 1, a score of 0.5 commonly indicates that model performance would be less accurate than random chance. The range of habitat suitability scores is between 0 and 1, but a score of 0.5 does not indicate that the likelihood of An. albimanus occurring there is neutral, or no better than any other area. It is simply a way to summarize the combination of environmental variables and pixel values, which include values well above 0.7 as shown in the boxplots of Figure 8, the differences being signficant according to Welch t tests presented in the figure legend. Typically, we consider a habitat suitability score of 0.0 - 0.3 to be poor, 0.3 - 0.5 to be moderate, 0.5 - 0.7 to be good, and 0.7 - 1 to be high suitability. For further reference please check "Habitat Suitability and Distribution Models with application in R" by Guisan et al 2017.
3) The environmental rasters were collected for the entirety of Costa Rica. We did not use any new data when examining plantations and the malaria outbreaks. We simply used the already created SDM and examined its projections in these particular areas. This is specifically described in lines 327-333 in the methods "As a proof of concept, we compared An. albimanus habitat suitability, measured as a probability, in land used for palm (Figure 2A) and pineapple (Figure 2B) plantations with that of the remaining land in the plantation districts. Based on estimates for An. albimanus dispersal, which has been recorded occurring up to 3 km [83, 84], we also compared the suitability in the plantations plus buffers of 1, 2 and 3 km with that of the remaining, plantation surrounding, land in the plantation districts (Figure 3)."
4) There are likely other potential reasons for the outbreak, but this doesn’t necessarily mean that they are related to habitat suitability for An. albimanus. We considered the outbreak because it is relevant to the vector and we were interested in vector presence in these particular areas. However, if our ultimate goal was to understand what led to the outbreak, it would be more useful for us to model habitat suitability for malaria - in which case, vector presence would only be a contributing variable. While you certainly need the vector to have a disease outbreak, there are many other factors that may influence the likelihood of an outbreak. More vectors does not uniformly translate into more disease. That said, it would be very nice to include the variables you mentioned above, but collecting this type of data would be difficult as the number of water puddles and intensity and type of mosquito control are inconsistent across time and space. However, these considerations could be addressed in a future project using a different method of analysis. In the discussion we actually hypothesize which vectors could have been in the spots with low habitat suitability. See lines 580-586 in the discussion: "Regarding, other potential vector species present in the Crucitas open-pit gold mine, possibilities include Anopheles vestitipennis Dyar & Knab and Anopheles punctimacula Dyar & Knab as these species thrive in recently disturbed environments and commonly co-occur with An. albimanus in Mesoamerica [106, 107]. Similarly, Anopheles darlingi Root, has been predicted to be present in the area with SDMs [108], has been reported in Panamá [109], but has not yet been detected in Costa Rica."
Minor comments
Figure 1: Some positive areas are represented by one record location only, whereas other positive areas have multiple location points. Did this cause any overrepresenting/underrepresenting bias in the modeling, how did you balance the potential bias?
The analysis is based on pixels, not on the districts. The districts with more locations simply represent that in those areas the mosquito was more widespread spatially. As explained in the methods we used the SRE algorithm to have a balanced number of pseudo-absences which are necessary for a sound SDM analysis based on the techniques we employed for our ensemble. Please refer to the methods in lines 153-193 in the manuscript for full details.
L199: what about the fine resolution images? Given that habitats of mosquitoes, especially larvae are very small, using fine resolution images may help improve the models.
This study is pretty innovative given the 250m resolution when compared with previous models whose resolution was above 1 km.
L292: Why choosing an AUC score of 0.7 as the inclusion criterium? “was fit” or “was fitted”?
Because is a standard threshold used in SDM studies that provides a good cut off in ROC analysis. This is standard in the SDM literature and does not normally get reported.
Figure 5: why NVMA was included in the lower model given the upper model showed its importance smaller than 5%?
Because the confidence limits included 5% (blue vertical lines).
L412: rainfall may have a significant impact on mosquito occurrence when it is limiting the mosquito life history. As mentioned above, consider adding data of lower rainfall (< 1000 mm) to enhance the generality of modeling.
Values below 1000 mm were included in the analysis as shown in figure 7.
L456: A low HS in malaria transmission areas deserves more details and discussion, including the possibility of missing important predictors, such as number of water puddles and intensity of mosquito interventions.
We discussed the possible presence of other vectors. Also note this study is about SDMs with a 250m resolution which is large when compared with larval habitats.
Reviewer 2 Report
Suggestions and comments to manuscript “Anopheles albimanus (Diptera: Culicidae) ensemble distribution modeling: applications for malaria elimination”
This is a very interesting MS on the use of a predictive model for Anopheles albimanus populations, a very important vector of Plasmodium sp. in Latin America. This research has been carried out by a large multidisciplinary team, which is reflected in the quality of the MS. I believe that the contribution to predicting the occurrence of An. albimanus populations by associating habitat suitability with higher order measures of variability in environmental variables is quite relevant.
I also like the inclusion of a social component in the Discussion of this MS, which shows how that knowledge generated by the biological sciences actually supports the social sciences, increasing their positive impact on people in general.
I only have two suggestions: On lines 317 and 336, italicize both "Anopheles albimanus". I think your Introduction is very complete, but I think it would be richer if you included some related classic studies by Rodríguez et al., from the Centro de Investigación de Paludismo (Malaria Research Center), in Tapachula, Chiapas Mexico.
Author Response
We thank the reviewer by their kind comments and appreciation of the plantationocene discussion.
1) We italicized the text as requested (the original was in italics but seems the editorial office corrected it).
2)We also included two additional references to the work from Chiapas (note ref 105 in the previous ms version [107 in the current] already was from the Chiapas malaria research group).
We added references 94 and 95 from the Chiapas studies:
- Rodriguez AD, Rodriguez MH, Hernandez JE, Dister SW, Beck LR, Rejmankova E, et al. Landscape surrounding human settlements and Anopheles albimanus (Diptera: Culicidae) abundance in Southern Chiapas, Mexico. J Med Entomol. 1996;33(1):39-48. Epub 1996/01/01. doi: 10.1093/jmedent/33.1.39. PubMed PMID: 8906903.
- Hernandez JE, Epstein LD, Rodriguez MH, Rodriguez AD, Rejmankova E, Roberts DR. Use of generalized regression tree models to characterize vegetation favoring Anopheles albimanus breeding. J Am Mosq Control Assoc. 1997;13(1):28-34. Epub 1997/03/01. PubMed PMID: 9152872.
Reviewer 3 Report
This paper was an interesting exploration of different species distribution models and their performance in predicting An albimanus, followed up by correlation with observed malaria outbreak locations. In general, I think this is a well-written manuscript that needs some clarifications, but no major methodological changes.
In the intro, I think the ENM vs SDM difference could be better described. They seem fairly similar to me, but the authors describe "coordinates" vs "environmental gradients" as the difference--does this mean SDMs predict onto a set of points while ENM predict onto gridded surfaces? If this is the difference, it remains unclear to me why (line 86-87) "[SDMs] are relevant for testing broad biogeographic/evolutionary hypotheses, while [ENMs] are useful for testing the transferability of niche models in space". It's also unclear to me exactly why SDMs overcome extent of occurrence issues. I think the first two paragraphs could be expanded somewhat to address this.
In the materials and methods, the time periods that An albimanus occurrence data came from should be mentioned. This is especially important because the authors emphasize the importance of using SDMs in the context of rapidly-changing land use patterns (which I agree with), but that implies only recent data would be used, ideally. The covariate data implies the earliest data were from 2000, but it would be good to be explicit.
Alongside this, I think it would be worth addressing the timeliness of the data/model directly. Do you get different results if you subset to only recent years? If there's too little data to do this (I expect so), then some text in the Discussion about this as a potential limitation in accounting for land use change would suffice.
Generally, the results are well-described and the comparisons between models was interesting. However, I think the utility of an ensemble model needs to be justified quantitatively, and requires some extra text/analysis. Did it actually do better than the constituent models? How much better was AUC/TSS with the ensemble model? How much did the results differ with the ensemble model against the constituent models?
Minor comments:
Figure 8 could be simplified significantly, such that the x-axis is "spatial buffer size", the y axis stays the same, and for each spatial buffer size you have two box and whiskers for non-plantation and plantation. Then it becomes a figure with only two subplots, one for Palms and another for Pineapples.
The relationship of B/C/D to A in Figure 9 is a little confusing. B seems like it should be the area represented by a square in A, but I don't see any dark red in that region in the A, where there clearly is in B?
In the discussion, the authors mention that "the mosquito was likely present at the two main towns also affected by the epidemic". This would be nice to see cited--and on the maps, it would be nice to see towns depicted as well, just to better orient the reader.
Author Response
This paper was an interesting exploration of different species distribution models and their performance in predicting An albimanus, followed up by correlation with observed malaria outbreak locations. In general, I think this is a well-written manuscript that needs some clarifications, but no major methodological changes.
We thank the reviewer for the appreciation of the work. In the next lines we describe the changes made to the manuscript.
In the intro, I think the ENM vs SDM difference could be better described. They seem fairly similar to me,
They are, but the SDM is the espatially explicit as explained in line 76.
but the authors describe "coordinates" vs "environmental gradients" as the difference--does this mean SDMs predict onto a set of points while ENM predict onto gridded surfaces? If this is the difference, it remains unclear to me why (line 86-87) "[SDMs] are relevant for testing broad biogeographic/evolutionary hypotheses, while [ENMs] are useful for testing the transferability of niche models in space". It's also unclear to me exactly why SDMs overcome extent of occurrence issues. I think the first two paragraphs could be expanded somewhat to address this.
The text was modified as follows: "In other words, SDMs overcome the limitations of traditional approaches such as the widely implemented “Extents of Occurrence” [1, 8, 9] for depicting the spatial range of a species as they are not based on opinions but quantitative relations. " clarifying how SDMs are more of a quantitative approach.
In the materials and methods, the time periods that An albimanus occurrence data came from should be mentioned. This is especially important because the authors emphasize the importance of using SDMs in the context of rapidly-changing land use patterns (which I agree with), but that implies only recent data would be used, ideally. The covariate data implies the earliest data were from 2000, but it would be good to be explicit.
We now clarified the bulk of the observations (from the entomological surveillance and the genetic studies were collected after 2000). See lines 163-164:"Records from the genetic studies and the vector control program were collected after 2000. "
Alongside this, I think it would be worth addressing the timeliness of the data/model directly. Do you get different results if you subset to only recent years? If there's too little data to do this (I expect so), then some text in the Discussion about this as a potential limitation in accounting for land use change would suffice.
This is outside the goals of the manuscript [see lines 144-151: "Here we use mid-resolution spatial data, 250 m, where we incorporate several layers derived from remotely-sensed and locally-measured environmental variables to create an ensemble SDM for An. albimanus. This SDM, that combines predictions from an ensemble of several quantitative methodologies, is a robust approximation to the distribution of this major malaria vector which we use to retrospectively assess the possibility this vector was present in transmission foci associated with malaria epidemics in 2019 [31, 38] and in landscapes used for pineapple production, where some malaria outbreaks have been recurrently observed over recent years."] and actually missguiding about illustrating the point about measuring higher orders of variability in the environmental variables, which require repeated obervations through time. Moreover, also ignores basic concept about the definition of climates as deriving from observations over extended periods of time (for example, see Strahler textbook on Physical Geography). Thus, we prefer not to change the text nor add discusssion that will keep the manuscript of track. If the question is worth exploring, the data are available in the repository indicated in the manuscript text.
Generally, the results are well-described and the comparisons between models was interesting. However, I think the utility of an ensemble model needs to be justified quantitatively, and requires some extra text/analysis. Did it actually do better than the constituent models? How much better was AUC/TSS with the ensemble model? How much did the results differ with the ensemble model against the constituent models?
This information is normally not reported as the values tend to be too high by the way information from the runs is merged based on AUC values. The exercise is tautological (mathematically the ensemble can not be worse than any singular model run), but to satisfy the reviewer question we now report the values in the legend of figure 6 and highlight what is more than a established expectation. See lines 389-391 "The AUC (mean ± SD) for this ensemble model was 0.983 ± 0.001 and the TSS (mean ± SD) was 0.833 ± 0.007 values outperforming all individual models presented in Table 2." We don't discuss the point as the expectation was set up in the intro, and has been more than discussed in the literature.
Minor comments:
Figure 8 could be simplified significantly, such that the x-axis is "spatial buffer size", the y axis stays the same, and for each spatial buffer size you have two box and whiskers for non-plantation and plantation. Then it becomes a figure with only two subplots, one for Palms and another for Pineapples.
We prefer to keep it as it as for presentation purposes or dicussion some people might actually want to focus on a single subplot.
The relationship of B/C/D to A in Figure 9 is a little confusing. B seems like it should be the area represented by a square in A, but I don't see any dark red in that region in the A, where there clearly is in B?
That is what happens when high resolution figures get amplified. Due to pixel compression the geometries at a higher scale can show details not observable at lower scales. A good reference for this is the book on "Fractals: a users guide ..." by George Sugihara and Harold Hastings. Colorwise, panel d is darker just as observed for Boca Arenal in panel A, and panel B (Crucitas) is incomplete as the radius goes across the Costa Rican border.
In the discussion, the authors mention that "the mosquito was likely present at the two main towns also affected by the epidemic". This would be nice to see cited--and on the maps, it would be nice to see towns depicted as well, just to better orient the reader.
The maps would become cluttered by adding additional vectorial information that ultimately is not informative. The suggestion is outside the standards to do understandable maps as discussed in basic GIS textbooks like the one by O'Sullivan and Unwin (Geographic Information Analysis) or the one by Brunsdom and Comber (An Introduction to R for spatial analysis and mapping). Also Note Table 3 summarizes what figure 9 depicts in a more graphical form. We present the info in those two ways so that readers can relate the pixel information with its statistical summary.
Round 2
Reviewer 1 Report
Authors have done a great job clarifying the issues. However, some important information that was mentioned in their response should also be added in the main text for clarity. For example, authors mentioned their model “was created specifically for Costa Rica” and “While we could apply this niche model to other countries/scenarios it would be careless to do so without some revisions to the model”. Authors should consider adding “in Costa Rica” in their title: Anopheles albimanus (Diptera: Culicidae) ensemble distribution Modeling in Costa Rica: applications for malaria elimination. Secondly, authors explained why they chose 0.7 as the standard score for AUS and how different HS scores indicate different model performance in their response. I think this information would be valuable for readers, especially those who are not familiar their models, to understand the robustness/goodness of their modeling.
Author Response
Authors should consider adding “in Costa Rica” in their title: Anopheles albimanus (Diptera: Culicidae) ensemble distribution Modeling in Costa Rica: applications for malaria elimination.- Thank you for your comment. We prefer to keep the current title as it emphasizes the use of ensemble models for malaria elimination context. Moreover, adding ¨in Costa Rica¨ would unnecessarily lengthen the title. Please, also note Costa Rica is right away mentioned in the abstract, summary and as a key-word, a reason why we really don t see any need to include "Costa Rica" in the title.
- We only consider AUC scores of 0.7 and above as the literature indicates such values are considered to demonstrate high model performance.
- Added an explanation at the end of section 2.6 in the methods section.
- Added [see lines 296-298] "We only consider AUC scores of 0.7 and above as they are considered to demonstrate high model performance [75]. ".
- We also highlighted this point about AUCs in the results section.
- Added [see lines 394-397]: "Habitat suitability ranged across Costa Rica from 0 to 1, with a score of 1 representing a habitat where An. albimanus should be present. There is some variation in classification of these scores, but we consider scores ranging from 0 – 0.3 to have poor suitability, 0.3 – 0.5 to be moderately suitable, 0.5 – 0.7 to have good habitat suitability, and 0.7 – 1 to be a highly suitable environment [75]."